# Role of GSH and Iron-Sulfur Glutaredoxins in Iron Metabolism—Review

**DOI:** 10.3390/molecules25173860

**Published:** 2020-08-25

**Authors:** Trnka Daniel, Hossain Md Faruq, Jordt Laura Magdalena, Gellert Manuela, Lillig Christopher Horst

**Affiliations:** 1Institute for Medical Biochemistry and Molecular Biology, University Medicine, University of Greifswald, 17475 Greifswald, Germany; danieltrnka@gmx.de (T.D.); mohammed.hossain@uni-greifswald.de (H.M.F.); L.Jordt@gmx.de (J.L.M.); manuela@gellert.org (G.M.); 2Christopher Horst Lillig, Institute for Medical Biochemistry and Molecular Biology, University Medicine Greifswald, Ferdinand-Sauerbruch-Straße, 17475 Greifswald, Germany

**Keywords:** glutathione, glutaredoxin, iron-sulfur cluster, iron

## Abstract

Glutathione (GSH) was initially identified and characterized for its redox properties and later for its contributions to detoxification reactions. Over the past decade, however, the essential contributions of glutathione to cellular iron metabolism have come more and more into focus. GSH is indispensable in mitochondrial iron-sulfur (FeS) cluster biosynthesis, primarily by co-ligating FeS clusters as a cofactor of the CGFS-type (class II) glutaredoxins (Grxs). GSH is required for the export of the yet to be defined FeS precursor from the mitochondria to the cytosol. In the cytosol, it is an essential cofactor, again of the multi-domain CGFS-type Grxs, master players in cellular iron and FeS trafficking. In this review, we summarize the recent advances and progress in this field. The most urgent open questions are discussed, such as the role of GSH in the export of FeS precursors from mitochondria, the physiological roles of the CGFS-type Grx interactions with BolA-like proteins and the cluster transfer between Grxs and recipient proteins.

## 1. Introduction

Glutathione, the γ-l-glutamyl-l-cysteinyl-glycine tri-peptide, is a ubiquitous nucleophile required in redox homeostasis, detoxification, and iron homeostasis [1]. Since the reactivity of glutathione (GSH) itself with proteins, small molecules, and xenobiotics is too low to be significant in vivo, see for instance, [2], GSH-dependent reactions need to be catalyzed by enzymes. These enzymes include glutaredoxins (Grxs), glutathione peroxidases (GPxs), glutathione reductase (GR), glutathione S-transferases (GSTs), and protein disulfide isomerases (PDIs) [1]. Nevertheless, the functions of GSH depend on the reactivity of its cysteinyl thiol group. Thiols can complex metals, be alkylated to thioethers, but they can also be oxidized to disulfides. In the case of glutathione, two molecules of reduced GSH can be oxidized to form glutathione disulfide (GSSG). Re-reduction is catalyzed by GR at the expense of NADPH. Being present in millimolar concentrations in most organisms, GSH was characterized as the “redox buffer” of the cell. In fact, the loss of GSH-utilizing enzymes may result in disrupted redox homeostasis, as in the case for GPxs [3,4], with effects as dramatic as cell death by a process named ferroptosis induced by the lack of GPx4 activity [5,6,7]. The loss of glutathione itself, however, firstly results in defects in cellular iron homeostasis [8,9]. The enzymes that catalyze or mediate most glutathione functions in iron metabolism are the iron-sulfur cluster (FeS)-containing Grxs.

In this review, we will address the functions of both GSH and FeS-Grxs in iron metabolism. This topic has been addressed before, see for instance [9,10,11,12]. The focus of this review is on mammalian cells, however, we take into account other taxa when discussing ground-breaking and major findings, both to illustrate the high degree of conservation in the systems as well as some unique aspects.

## 2. Glutathione, Glutaredoxins, and Iron-Metabolism

As early as 1972, Tanaka and coworkers reported that, in vitro, GSH may be able to complex iron, resulting in absorption spectra resembling those of FeS clusters [13]. It took nearly two decades before the essential function of GSH in the synthesis and maturation of FeS cluster proteins in vivo was discovered in yeast. Defects in the biosynthesis of FeS proteins in *Saccharomyces cerevisiae* are associated with a more than two-fold increase of GSH [14]. However, this increase was apparently not caused by concomitant disturbances in redox metabolism. In fact, the depletion of GSH impaired the maturation of FeS proteins substantially, most of all affecting the non-mitochondrial FeS proteins [15].

### 2.1. Iron-Sulfur Cluster Biogenesis

FeS clusters are essential for life. They participate in the transfer of electrons, are cofactors in enzymatic catalysis, control the stability of biomolecules, and act as regulatory elements [16,17,18]. Mitochondria are essential for FeS cluster biogenesis. Not only are all the FeS clusters required for energy conversion synthesized here, the maturation of cytosolic FeS clusters also depends on a yet to be defined mitochondrial precursor [19,20]. Mitochondrial FeS cluster biogenesis and the cluster transfer to target apo-proteins have been studied to great detail and reviewed comprehensively before, e.g., in [21,22,23]. In brief, Fe_2_S_2_ centers are synthesized from iron and cysteine-derived sulfur by the early iron-sulfur cluster (ISC) synthesis machinery. From there, the Fe_2_S_2_ centers are likely transferred to the monothiol Grx5 with the aid of a heat shock protein of 70 kDa (Hsp70) chaperone system, the details of which will be discussed below. In the present models, Grx5 acts as a hub, from which the FeS centers are (1) either directly, or with the help of additional “targeting factors”, transferred to Fe_2_S_2_ target proteins or (2) transferred to a protein complex composed of the assembly factors ISCA1, ISCA2, and IBA57, where they are combined to Fe_4_S_4_ centers [24]. Only recently, Lill and co-workers succeeded in biochemically reconstituting the assembly of Fe_4_S_4_ clusters [25]. The process requires the ISC machinery, holo-Grx5, and reduced Fdx2 for the reductive fusion of two Fe_2_S_2_ clusters into one Fe_4_S_4_ cluster. These clusters are trafficked by ISC-targeting factors, such as proteins from the BolA family and iron-sulfur cluster scaffold homolog 1 (NFU1) to apo-target proteins such as the mitochondrial aconitase (ACO2). The maturation of the Fe_4_S_4_ centers in the complex I (NADH-ubiquinone oxidoreductase, NDUF) subunits NDUFS1, NDUFV1, and NDUFS2, 7, and 8 seems to require the P-loop NTPase Ind1 (iron–sulfur protein required for NADH-dehydrogenase) [26]. The molecular mechanisms of cluster insertion and the recipient protein pre-requisites are still unclear.

The insertion of FeS proteins into apo-target proteins requires their cysteinyl sulfur ligands to be in the reduced thiol state. Under normal conditions in yeast mitochondria, this seems to be independent of the major thiol reducing systems, i.e., the GSH/Grx and thioredoxin (Trx) systems [27]. However, for mammalian cells and proteins, a number of conditions have been characterized that lead to the oxidation of such cysteinyl residues, for instance, in complex I subunits NDUFS1 and NDUFV1 in a murine Parkinson’s disease model as a consequence of GSH depletion [28]. In general, GSH appears to be a crucial factor for complex I activity, especially in neurons [29,30,31].

From groundbreaking work in yeast cells, it was established that cytosolic FeS biogenesis depends on a sulfur compound generated inside mitochondria [32]. This not yet specified sulfur compound (X-S) must be exported as source for cytosolic FeS cluster synthesis. This compound must be transported into the cytosol in an ATP-dependent manner. The ATP-binding cassette (ABC) transporter Atm1 (in human ABCB7), located in the inner mitochondrial membrane, soon came into focus as the prime candidate for this function [32]. Atm1/ABCB7 may also be the link for the necessity of GSH for the maturation of cytosolic FeS proteins. Structural analyses revealed a large cavity in the dimeric Atm1 close to the inner membrane surface that can accommodate GSH [33,34,35]. In fact, the size of the cavity also allows for the binding of GSH as part of a larger not yet defined substrate transported by Atm1 [34]. Mutations in the binding site of ABCB7 that inhibit binding of a GSH moiety result in decreased functions of cytosolic FeS proteins and mitochondrial iron overload [33,36,37]. The same phenotype is caused by GSH or Atm1 depletion in yeast [15]. Atm1 and the *Arabidopsis thaliana* homolog ATM3 transport GSSG with increased ATPase activity but neither Fe^2+^ nor reduced GSH stimulate the ATPase activity of the transporter [38]. Atm1 also transports GS-S-SG (glutathione trisulfide) [38]. Since substantial data suggest that only a sulfur compound is required as the mitochondrial contribution to cytosolic FeS biogenesis (for a summary, see [34]), such per- or poly-sulfides are compelling candidates as endogenous substrates. More recently, however, intact Fe_2_S_2_ clusters ligated by four GSH molecules have also been suggested as Atm1 substrate [39]. This is supported by recent kinetic studies that reported the transport of (GSH)_4_Fe_2_S_2_ by Atm1 with a ~100-fold increased activity in the presence of the cluster [40]. The open questions, in addition to the nature of the transported compound, are the generation of this GSH derivative(s) from the Grx5/GSH-ligated Fe_2_S_2_ cluster in mitochondria (see below) as well as the link to the cytosolic iron sulfur cluster assembly machinery (CIA). In vitro, the GSH-ligated Fe_2_S_2_ cluster itself can be used to reconstitute apo-proteins [41].

The biogenesis of cytosolic and nuclear FeS-proteins requires numerous proteins and is facilitated by the CIA, summarized in [42]. Overall, the process can be divided into two steps. First, a Fe_4_S_4_ cluster is assembled on a scaffold complex. Second, this cluster is transferred and inserted into recipient apo-proteins. Apart from the yet to be determined component X-S that has to be supplied by the mitochondrial FeS synthesis machinery, the cytosolic FeS protein biogenesis requires an iron source and the supply of electrons. The latter are supplied by the NADPH-dependent diflavin oxidoreductase 1 (NDOR1) and the FeS protein anamorsin/CIAPIN1 (see Figure 1) [43,44,45,46]. The first step is the assembly of the Fe_4_S_4_ cluster on the cytosolic FeS scaffold complex (see Figure 1) [47,48,49]. The heterotetrameric complex consists of two P-loop NTPases, CFD1 (cytosolic FeS cluster-deficient protein 1) and NBP35 (nucleotide-binding protein 35). The complex binds two Fe_4_S_4_ clusters, one at the *N*-terminus of NBP35 and the second one is transiently ligated between CFD1 and NBP35 [50]. The nature and source of the iron for the formation of this first cytosolic FeS cluster is still unclear. Multi-domain Grxs like yeast or mammalian Grx3 may be potential candidates. They do play a general role in iron trafficking through the cytoplasm in a GSH-dependent manner [51,52] (details below). Potentially, they function in concert with a BolA family protein [10,53,54,55,56]. In the second step, the primarily assembled cluster is transferred and inserted into target proteins. This is facilitated by IOP1 (iron-only hydrogenase-like protein) and the CIA-targeting complex [57]. IOP1 may acts as a CIA adapter protein, mediating the contact between early and late parts of the CIA machinery, although the exact mechanism is still under discussion (see Figure 1) [47,58,59]. Probably, cytosolic iron-sulfur protein assembly protein 1 (CIA1), CIA2b, and MMS19 (MMS19 nucleotide excision repair protein homolog) build variable hetero-complexes, called the CIA-targeting complexes, that directly interact with specific cytosolic and nuclear apo-FeS proteins (see Figure 1) [58,60].

### 2.2. Glutaredoxins

Grxs form a branch of the Trx family, for an overview see [61]. Bacterial Grxs represent the most basic representation of the Trx-fold, consisting of a four-stranded central β-sheet surrounded by three α-helices, and Grxs of higher organisms frequently display additional N- and C-terminal helices (Figure 2). In 1976, the first Grx was defined as a GSH-dependent electron donor for ribonucleotide reductase (RNR) and thus DNA synthesis [62]. In the following years, Grxs were comprehensively characterized as oxidoreductases that catalyze the formation and reduction of disulfides, i.e., inter- and intra-molecular protein disulfides, and with high specificity disulfides between protein thiols and GSH, i.e., reversible (de-)glutathionylation. For comprehensive reviews on this topic, see for instance [61,63,64,65,66,67], and for a summary of the characteristics of the human Grxs, see Table 1. In brief, these redox-active Grxs (CPYC-type or class I Grxs) contain a consensus Cys-Pro-Tyr-Cys active site motif and catalyze thiol-disulfide exchange reactions in two connected reaction mechanisms. The formation and reduction of protein disulfides require both active site cysteinyl residues and (de-)glutathionylation of only the more N-terminal. The mechanisms were thus termed dithiol and monothiol reaction mechanisms. Both reactions are initiated by a nucleophilic attack of the more N-terminal cysteinyl residue, which is characterized by a particularly low pK_a_ value ≤ 5 [66,68,69,70], on the target disulfide. In the case of the dithiol reaction mechanism, the intermediate disulfide between the Grx and the target protein is reduced by the more C-terminal cysteinyl residue. The monothiol mechanism results in a reduced protein and a disulfide between the Grx and GSH (see Figure 2). Reduction of the Grx with a disulfide in the active site by GSH results in the same Grx-GSH mixed disulfide, which can be reduced by another molecule of GSH, completing both reaction cycles. Both reactions are fully reversible, as Grxs catalyze both the oxidation and reduction of target proteins. A second class of Grxs came into focus much later. These proteins share the consensus active site motif Cys-Gly-Phe-Ser, hence CGFS-type or class II Grxs. With few exceptions [71,72], CGFS-type Grxs are inactive as oxidoreductases. Instead, these proteins function in cellular iron metabolism [10,51,52,73,74], see below.

For decades, Grxs were characterized as co-factorless oxidoreductases [67]. It therefore came as a big surprise when the first two FeS-Grxs were described, *Arabidopsis thaliana* GrxC1 [83,84] and human Grx2 [16,85]. In both cases, it turned out that the exchange of the prolyl residue in the CPYC consensus active site for a glycyl and seryl residue, respectively, was sufficient to allow cluster ligation [84,85]. The second big surprise was the mode of cluster ligation itself in these proteins. The clusters are ligated in a dimeric holo-complex at the interface of two hardly interacting Grx monomers [84,86]. The [Fe_2_S_2_]^2+^ clusters are ligated by the two more N-terminal cysteinyl residues of the active site and the thiol groups of two non-covalent bound GSH molecules [84,85,86]. These were the first examples of FeS clusters co-ligated by GSH. Following these two C(non-P)YC-type Grxs, all CGFS-type Grxs have been characterized as Fe_2_S_2_-proteins, see for instance [10,74,80,81,87]. Both Grx sub-families bind the FeS cluster in a very similar way at the interface of the dimeric holo-complex, including co-ligation by GSH. However, one particular feature separates the two groups: the relative orientation of the Grx monomers in the holo-complex towards each other (Figure 3). Compared to the CGFS-type Grxs, the position of one monomer in the C(non-P)YC-type Grxs is tilted by approximately 90° toward the site relative to the other monomer. The sequestration of the N-terminal active site cysteinyl residue in the holo-complex of the redox-active, yet FeS-binding, Grxs suggests that the cluster serves as a regulatory mechanism controlling the activity of the proteins, for instance, by increased levels of GSSG or nitrogen oxide (NO) [16,88], or that it may serve other redox-independent functions [16,89].

Although both types of Grxs discussed here share highly similar 3-D structures (see Figure 3), as well as all elements and residues required to bind GSH [9], they exhibit completely different activities—oxidoreductase versus transferase. The mechanistic basis for this profound difference was the subject of many investigations and speculations [81,87,90,91], until two studies recently characterized the molecular basis of their distinct activity profiles [92,93]. In brief, the key determinants of their function are unique loop structures just before the active site. The engineering of a CxxC-type Grx with a CGFS-type loop switched its function from oxidoreductase to FeS transferase in a zebrafish model and the introduction of a CxxC-type loop into a CGFS-type Grx abolished its FeS transferase activity and activated the oxidative half-reaction (Figure 2, reaction 5 reverse) of the oxidoreductase [92]. The reductive half-reaction, requiring the interaction with the second GSH molecule (Figure 2, reaction 4), is dependent on further elements, characterized in detail in [93]. Together, these studies explain how subtle structural differences determine the diverse Grx functions. An overview of the different classes of Grxs in different species is depicted in Figure 4.

#### 2.2.1. Vertebrate- and Mammalian-Specific Glutaredoxin 2

Before it was described as an FeS protein, human Grx2 was characterized as a redox-active Grx with the ability to reduce mixed disulfides and effectively (de-)glutathionylate target proteins [79,94,95]. The human GLRX2 gene consists of five exons, including two alternative first exons (Ia and Ib) leading to three transcript variants. The core domain of Grx2, including the active site, is encoded by exon II-IV. GLRX2_v1 (exon Ia-II-III-IV) encodes the ubiquitously expressed Grx2a, including a mitochondrial targeting sequence. GLRX2_v2 and v3 are products of the alternative splice donor sites of exon Ib, encoding the nuclear and cytosolic isoforms Grx2b and Grx2c. The expression of Grx2b and Grx2c is restricted to the testis in adult human tissues, but has also been demonstrated in various cancer cell lines [77]. In contrast, the mouse GLRX2 gene consists of six exons, three constitutive exons (II, III, IV), two alternative first exons (Ia, Ib), and one single cassette exon. Five transcript variants encode three protein isoforms. The mitochondrial Grx2a and the nuclear/cytosolic Grx2c are conserved from mouse to human. Testis-specific Grx2d is unique to mouse [96]. Grx2 shares 34% sequence homology with Grx1 and a CSYC active site motif, with the exchange of the prolyl for a seryl residue [79,94]. This altered active site sequence results in an increased affinity for glutathionylated proteins and it can be reduced by either GSH or thioredoxin reductase, combining characteristics of Trxs and Grxs. Dimeric inactive holo-Grx2 bridges an FeS cluster. Degradation of the cluster in oxidative conditions, e.g., a more oxidized glutathione pool, results in monomeric active Grx2, indicating a function as a redox sensor in vivo [16]. Monomerization and cluster disassembly can cause lipid peroxidation, a drop in mitochondrial membrane potential, and eventually cell death [97]. The mitochondrial Grx2a was shown to participate in the maintenance of the redox equilibrium under conditions that promote oxidative damage to mitochondrial proteins. Especially for cells over-expressing Grx2a, protective functions have been described [88,98,99]. Grx2a over-expression decreased susceptibility towards apoptosis induced by doxorubicin (DOX) [100]. Grx2 is essential for mitochondrial morphology and dynamics in cardiomyocytes in humans and mice [101]. The loss of mitochondrial Grx2 is connected to increased mitochondrial proton leaks and respiration in muscle cells [102]. Genomes of other vertebrate species, e.g., zebrafish, contain genes encoding homologs to the cytosolic Grx2 isoform. This cytosolic zfGrx2 is essential for brain development. Zebrafish with silenced expression of cytosolic Grx2 lose essentially all types of neurons by apoptotic cell death and fail to develop an axonal scaffold. Only the re-introduction of wildtype Grx2c could rescue the defects, but not in either of the redox-inactive active site mutants [103]. The over-expression of Grx2c in SH-SY5Y neuroblastoma cells during retinoic acid-induced differentiation increases axon length and the number of branching points by up to two-fold [103]. Cytosolic Grx2 also has an essential function for the vascular development and maintenance of cardiovascular function [104,105]. Zebrafish lacking cytosolic Grx2 have an impaired heart looping and defects in heart functionality due to a failed migration of cardiac neural crest cells [106]. This heart looping defect could be rescued by introduction of the active site mutant of zfGrx2 that is still able to catalyze monothiol mechanism reactions [106]. Grx2c also has an essential function in spermatogenesis, a process that includes the migration of spermatogenic cells through the close Sertoli cell formation [77]. Recent results indicate a correlation between Grx2c expression and cancer-specific survival in clear cell renal cell carcinoma patients [107]. In a proteomic approach, Schütte et al. were able to identify target proteins, e.g., collapsin response mediator protein (CRMP) 2, that undergoes thiol-disulfide exchange reactions catalyzed by Grx2 [108]. In models of Parkinson’s disease, the depletion of glutathione resulted in a dose-dependent Grx2 inhibition and, similar to gene silencing of Grx2, decreased iron incorporation into complex I and ACO2. The loss of Grx2 function also led to the activation of iron regulatory protein (IRP1), resulting in the increase in the iron uptake protein transferrin receptor, decreased levels of the iron storage protein ferritin, and mitochondrial iron accumulation. In the cytosol, the loss of Grx2 resembled iron starvation conditions.

#### 2.2.2. Glutaredoxin 5

Human Grx5 is one of the central proteins in the mitochondrial ISC machinery, as well as in cluster trafficking and, therefore, iron homeostasis [109,110]. In humans, the maturation of mitochondrial FeS cluster-containing proteins can be divided into different steps. First, the initial Fe_2_S_2_ cluster is assembled on the iron-sulfur cluster enzyme ISCU (ISCU2) and involves at least 17 characterized proteins [111]. The human ISCU2-M140I variant can overcome the loss of frataxin. However, this is not by restoring its function in cluster assembly, but rather by the acceleration of cluster transfer from ISCU2 to Grx5 [112]. The release and transfer of the Fe_2_S_2_ cluster is facilitated by Hsp70 chaperones [22]. Chaperone binding enhanced the ATP-dependent cluster transfer from *E. coli* IscU in vitro [113]. In *S. cerevisiae*, the ATPase activity of the chaperone increased by Isu1 binding but not by interaction with Grx5. The association of Isu1, Grx5, and the chaperone is required for cluster transfer from Isu1 to Grx5 [114]. A study published in 2018, however, contradicted these findings by showing a cluster transfer from ISCU to Grx5 only in the absence of the human mitochondrial Hsp70 chaperones HSPA9 and HSC20 [115]. However, this study completely relied on in vitro data with assay times up to two hours, and therefore the results were mainly subjected to thermodynamic restrictions rather than physiological constraints.

Together with Grx5, ISCA1 and ISCA2 are proteins involved in the assembly of Fe_4_S_4_ clusters [22]. Until recently, only very slow rates of cluster transfer from Grx5 to ISCA1 and ISCA2 were demonstrated in vitro. Although new insights were provided by the structure of the ISCA2-IBA57 complex [10,11], the reductive fusion of the two Fe_2_S_2_ clusters to one Fe_4_S_4_ cluster has only recently been reported [25]. BolA-like proteins, more precisely BolA1 and BolA3, were also suggested to interact with Grx5 in the assembly of Fe_4_S_4_ and possibly Fe_2_S_2_ clusters, as summarized, e.g., in [22]. Both human BolA1 and BolA3, interact with apo- and holo-Grx5 to form hetero-clusters with different affinities as shown in in vivo and in vitro studies [116,117]. NMR, EPR, CD, and UV/vis spectroscopy were utilized to characterize and identify differences in the nature of the clusters bound in the BolA1-Grx5 and BolA3-Grx5 hetero-complexes [118].

The loss of Grx5 disrupts FeS assembly on target proteins and leads to mitochondrial iron overload [19,114]. In *Schizosaccharomyces pombe*, Grx5 depletion also led to a decrease in mitochondrial DNA [119]. Additionally, in *S. cerevisiae*, an iron-dependent increased rRNA degradation was observed upon Grx5 depletion due to iron overload [120]. In zebrafish, a lack of Grx5 led to the activation of IRP1 and blocked heme biosynthesis [121]. The first step in this pathway is catalyzed by aminolaevulinate synthase 2 (ALAS2). The over-expression of ALAS2 RNA without the iron response element regulated by IRP1 rescued the zebrafish embryos, while the expression of ALAS2, including the iron response element, did not [121]. Human patients with decreased levels of Grx5 develop iron overload and sideroblastic-like microcytic anemia [82,109].

The mechanisms of cluster transfer in the mitochondrial FeS cluster machinery, as well as to target proteins, remain to be revealed. Over the years, many studies have been published proposing cluster transfer between proteins that are clearly involved in the mitochondrial FeS cluster synthesis pathway, but they have relied solely on in vitro data (e.g., [115]). As mentioned above, these sorts of in vitro studies are restricted by thermodynamics and do not take physiological conditions nor enzymatic catalysis into account. An example of how this leads in an unavailing direction can be found for CSYC-type Grx2. Also located in the mitochondria, Grx2 complexes an FeS cluster [16]. In contrast to Grx5 depletion, the loss of Grx2 does not impair ISC biogenesis or transfer but leads to defects, e.g., in brain and heart development [103,106]. In vivo Grx2 and Grx5 display completely different functions in redox regulation and iron homeostasis, respectively. However, it was published that in vitro human Grx2 transferred its FeS cluster to human ferredoxin (Fdx1) (see Table 2 and Figure 5) with an apparent second-order rate constant of 1160 ± 200 M^−1^ min^−1^ [122]. An essential reaction that takes more than 60 min is far away from being physiological and would be inconsonant with the human lifespan.

#### 2.2.3. Multi-Domain Glutaredoxins, Glutaredoxin 3

Multi-domain Grxs are unique to eukaryotic cells. They consist of an N-terminal, normally redox inactive, and a Trx domain followed by one to three CGFS-type Grx domains, each of which can complex the GSH co-ligated Fe_2_S_2_ cluster in dimeric complexes. The yeast *Saccharomyces cerevisiae* expresses two closely related multi-domain Grxs containing a single CGFS-type Grx domain each, Grx3 and Grx4. Two major functions, both central for iron metabolism, were characterized for these proteins: (1) Grx3 and Grx4 have a central role in intra-cellular iron trafficking and sensing. The depletion of Grx3/4 specifically impaired all iron-requiring reactions in the cytosol, mitochondria, and nucleus, including the synthesis of FeS clusters, heme, and di-iron centers, such as in RNR (see Figure 1). The cells failed to insert iron into target proteins, as well as to deliver iron to mitochondria. Iron was simply not bio-available in the absence of the proteins [51] and (2) the availability of iron to form the FeS-bridged holo-complexes of Grx3 and 4 is used as sensor for the iron state of fungal cells. Extensive analyses of *S. cerevisiae* and *S. pombe* have uncovered unique mechanisms that control iron metabolism in different fungi, summarized recently in [11]. The common thread in these regulatory mechanisms are the multi-domain CGFS-type Grxs that interact, often together with BolA-type proteins (see below), with transcription factors dependent on the iron state of the cell, thus controlling the transcription of proteins and enzymes that take part in, or control, iron metabolism. For detailed discussion on this topic, see for instance [10,11,133,134].

Vertebrate-specific Grx3, also known as protein kinase C-interacting cousin of thioredoxin (PICOT), TXNL-2, and HUSSY-22, contains two C-terminal CGFS-type Grx domains [66,135,136]. Grx3 is ubiquitously expressed [80,137], the protein can complex two Fe_2_S_2_ clusters at the interfaces between the two CGFS-type Grx domains in a homo-dimeric holo-complex, and it binds iron in vivo [80]. The depletion of Grx3 in zebrafish embryos primarily affected hemoglobin maturation. The loss of Grx3 function did not affect globin biosynthesis, and instead heme did not mature [52]. This was likely caused by the loss of an essential FeS cluster in the enzyme ferrochelatase that catalyzes the final step in heme maturation, iron insertion [138]. Gene silencing of Grx3 in cells of human origin (HeLa cells) induced a phenotype resembling an iron starvation phenotype despite the sufficient bio-available iron. The protein levels of several cytosolic FeS proteins were altered, for instance, IRP1 and glutamine phosphoribosylpyrophosphate amidotransferase (GPAT). The protein levels of ferritin were decreased and the levels of the transferrin receptor increased, indicating the activation of IRP1. Apparently, the Grx3-depleted cells were unable to use iron efficiently, indicating a central role for Grx3 in iron metabolism [52] similar to the one described in yeast [51], i.e., a function in cellular iron trafficking. The molecular base of this function and how it relates to the observed defects in FeS protein maturation in the cytosol of eukaryotic cells is still unknown.

Human Grx3 was initially identified as an interaction partner of protein kinase C θ and is associated with various signaling pathways that lead to the activation of cells [136]. Grx3 is essential during development, the loss of Grx3 in mice resulted in embryonic death between E12.5 and E14.5, without apparent defects in organogenesis [139]. Grx3^−/−^ embryos did not exhibit obvious histological abnormalities, however, the embryos were reported to be of smaller body size and developed hemorrhages in the head [139,140]. It is noteworthy that the time point of embryonic death, E12.5, also marks the onset of definitive erythropoiesis in the fetal liver [141,142] and, from this point on, erythropoesis is the major iron-consuming process. Grx3 can protect from cardiac hypertrophy in animal models. Grx3 protein levels were increased in these models and heterozygous Grx3^+/−^ mice were more vulnerable to developing cardiac hypertrophy, in contrast to wildtype mice [139,140]. Disturbances in iron metabolism have also been linked to cardiac pathologies. For instance, in Friedreich’s ataxia patients, the (partial) loss of the FeS cluster biogenesis protein frataxin (Figure 1) causes mitochondrial iron overload and defects in mitochondrial FeS maturation, summarized in [23,143]. These defects frequently cause cardiomyopathy and cardiac hypertrophy [144]. To date, however, it is unclear whether the role of Grx3 in cardiac hypertrophy is connected to its role in iron metabolism.

### 2.3. Grxs and BolA-Like Proteins

Both genetic and biochemical evidence link BolA-like proteins to iron metabolism and to CGFS-type Grxs in particular [10,145]. It was hypothesized that both proteins interact in the transfer of FeS clusters to targeting complexes or recipient proteins. This is supported by a number of in vitro and structural studies [118]. Unlike the CGFS-type Grxs, BolA-like proteins from different species show a high degree of heterogeneity and a low degree of conservation, including some of the residues that were suggested to take part in the ligation of FeS clusters in both homo- and hetero-dimeric holo-complexes.

In *S. cerevisiae*, the regulation of iron metabolism by the transcription factors activator of iron transcription protein (Aft) 1 and Aft2 depends on the Grx3/Grx4 siblings and the proteins Fe repressor of activation (Fra) 1 and Fra2, for summaries, see [10,11]. Fra2 is BolA-like protein also known as BolA2. The Fe_2_S_2_ cluster in the hetero-dimeric complex between Grx3/4 and Fra2 is complexed by the Grx3/4 active site CGFS cysteinyl residue, a Fra2 histidyl residue, one GSH, and another ligand that is not a histidyl residue and remains elusive [146,147]. The conserved His103 residue is not required for hetero-dimer formation and cluster binding in vitro, but influences cluster stability [146]. In vitro studies described a cluster transfer between Grx3-Bol2 and Aft2, involving a ligand exchange mechanism and a specific protein–protein interaction that requires Aft2 Cys187 [148,149]. Cluster binding appeared to be more stable in the hetero-dimeric complex compared to Grx3/4 homo-dimers, although removal of this cluster did not disrupt the Grx3-Fra2 hetero-dimer, raising the question of whether it functions as an FeS scaffold or iron sensing protein [147]. In the proposed iron sensing mechanism in *S. pombe*, an FeS cluster is transferred from the transcription factor iron-sensing transcription factor 1 (Fep1) to Grx4-Fra2 in response to iron starvation, thereby activating gene expression to increase the intra-cellular iron pool [150].

The holo-complex of the human multi-domain CGFS-type Grx3 bridges two Fe_2_S_2_ clusters with four GSHs and its two conserved CGFS motifs [80]. In 2012, Li et al. demonstrated that human Grx3 forms a heterotrimeric complex with human BolA2 in vitro, and this was confirmed by Banci et al. [53,56]. As in yeast, cysteinyl and histidyl residues of Grx3 and BolA2, respectively, were proposed to be involved in cluster coordination [53,56]. In contrast to yeast, however, in vivo data supporting this interaction and a physiological role of this hetero-trimeric complex remain to be presented. In vitro data suggested more stable Fe_2_S_2_ clusters in the hetero-trimeric compared to the Grx3 homo-dimeric complexes, as observed in yeast. Nevertheless, a role of the BolA2-Grx3 complex in Fe_2_S_2_ cluster transfer in the cytosolic FeS protein maturation pathway was proposed [56]. Cluster transfer from homo-dimeric Grx3 to CIAPIN1 (also named anamorsin, see Figure 1 and Figure 5) was demonstrated, the specific interactions between the two were proposed as key mechanisms in anamorsin maturation [125]. However, the hetero-complex with BolA2 was also reported to be able to transfer both bridging Fe_2_S_2_ clusters to CIAPIN1/anamorsin in vitro, and thus a function as an FeS cluster transfer component in the cytosolic FeS protein biogenesis was suggested [56]. The siRNA-mediated silencing of Grx3 induces an iron starvation phenotype in HeLa cells [52]. However, the silencing of BolA2 expression not only failed to induce a similar phenotype, but the co-silencing of Grx3 and BolA2 rescued the iron starvation phenotype to some degree (unpublished own data). These results imply an antagonistic rather than joint function of cytosolic Grx3 and BolA2. This fragmentary puzzle of information and results remains to be solved.

Mitochondria of eukaryotic cells usually harbor the CGFS-type Grx5 and two BolA-like proteins, BolA1 and BolA3, both of which can form hetero FeS-bridged complexes with Grx5. Uzarska et al. demonstrated that human apo-Grx5 and BolA1 or BolA3 also specifically interact in chemical shift assays [114]. This interaction involves the location surrounding the invariant histidyl residue in the BolAs and the GSH-binding site in Grx5 [117]. Complex holo-models suggest that Grx5-BolA3 undergo significant structural rearrangement upon dimer formation and FeS cluster binding (Figure 6) [118]. In the loop connecting β-strand 1 and 2 of BolA3, the Cys 59 residue moves towards the invariant C-terminal His 96 and coordinates the Fe_2_S_2_ together with the active site and GSH thiols of Grx5. The complex with BolA1, on the other hand, seems to not involve structural re-arrangements and has a different orientation [118] (Figure 6).

In vivo and in vitro studies suggest specialized functions of yeast mitochondrial Bol1 and Bol3 in the same pathway, i.e., FeS protein maturation in mitochondria. The over-expression of Grx5 increases the BolA1 level but there is no effect on BolA3 [116]. BolA1 interacts with Grx5 via the Fe_2_S_2_ cluster, whereas BolA3 interacts with Nfu1 in the Fe_4_S_4_ cluster assembly, although the detailed mechanisms remain unsolved [116,117]. Yeast cells lacking BolA1 and BolA3 show defects in Fe_4_S_4_ enzymes, e.g., aconitase and lipoic acid synthase [116,151]. The interaction of the two human mitochondrial BolA-like proteins, BolA1 and BolA3, and Grx5 was characterized by a number of in vitro techniques [117]. Conserved histidyl residues (His102 in BolA1 and His96 in BolA3) and other potential cluster ligands, e.g., histidyl and cysteinyl residues in BolA1 and BolA3, respectively, are involved in hetero-dimeric cluster formation [10,117]. A reduced Rieske-type Fe_2_S_2_ cluster is coordinated by Grx5 and BolA1 with high affinity, whereas the oxidized, Fdx-like cluster of Grx5 and BolA3 is labile and BolA3 preferably interacts with Nfu1 [117,118]. Based solely on in vitro studies, Sen et al. concluded the contrary—a significant BolA3-Grx5 interaction and a weak BolA3-Nfu1 interaction [124]. Two BolA3-Grx5 hetero-complexes can transfer their Fe_2_S_2_ clusters to Nfu1 to form a Fe_4_S_4_ cluster in FeS protein maturation [126]. A role in cluster trafficking was ruled out for BolA1-Grx5 because of its structurally buried cluster and the lack of transfer efficiency to common Fe_2_S_2_ cluster acceptors, such as ferredoxins [123]. Patients with mutations in the Grx5 or the BolA3 gene suffer from variations of nonketotic hyperglycinemia with decreased lipoylation, likely caused by interruption of the cluster transfer pathway to the FeS protein lipoate synthase [110]. The physiological role of BolA1 and BolA3 in complex with Grx5 remains elusive. For more comprehensive summaries of the topic, we refer to [10,54,111].

### 2.4. FeS Cluster Transfer Reactions

CGFS-type Grxs and BolA-like proteins have been suggested to cooperate in the transfer of FeS clusters to target proteins and targeting protein complexes [53]. A number of studies, summarized in Table 2 and Figure 5, have addressed such transfer reactions in vitro, mostly utilizing differences in the absorption or circular dichroism of the holo-complexes of target and recipient proteins. In brief, the combined in vitro data on these cluster transfer reactions can be summarized as follows: it appears that Fe_2_S_2_ clusters can be transferred between most proteins that have the ability to ligate them, independent of the physiological significance of these interactions and phylogenetics. The reactions are reversible and seem to primarily follow thermodynamic constrains. So far, evidence for the requirement of any form of catalysis has only been demonstrated for the transfer of the Fe_2_S_2_ cluster built in the initial scaffold ISCU to the Grx5 homolog of *Azotobacter vinelandii* in the form of the HscA/HscB chaperone system [131]. The rate constants, especially of the reactions regarded as physiologically significant, are generally low, in the range of 0.1–1.7·10^1^ M^−1^ s^−1^. Astonishingly, some of the highest rate constants have been reported for cluster transfer reactions between proteins that cannot be considered physiologically meaningful, e.g., from human Nfu (mitochondrial) to *S. cerevisiae* Grx3 (cytosolic) with 6·10^2^ M^−1^ s^−1^ [122], or from *S. cerevisiae* Grx3 to *Azotobacter vinelandii* ISCA at ≥8.3·10^2^ M^−1^ s^−1^ [130]. In vitro, human Grx2, a CSYC-type Grx, can transfer its cluster to Fdx1 with similar (low) rates as CGFS-type Grx5, i.e., 1.9·10^1^ M^−1^ s^−1^ and 3.3·10^1^ M^−1^ s^−1^, respectively, summarized in Table 2 and Figure 5. In vivo, however, due to its different quaternary structure, the ubiquitously expressed [77] mitochondrial Fe_2_S_2_-Grx2a cannot compensate for the iron deficiency phenotype caused by Fe_2_S_2_-Grx5 depletion or loss, in neither zebrafish [82] nor in humans [82,109]. With respect to the role of the BolA-like proteins, no in vitro study so far has demonstrated that these proteins are strictly required for the transfer of FeS clusters from Grxs to any other protein or vice versa, nor that their presence would enhance the rate constants of the transfer reaction. To date, in vitro studies have not provided conclusive evidence on the nature of the efficiency, nor the specificity in cluster transfer reactions observed in vivo.

## 3. Other Glutathione-Iron Complexes

Dinitrosyl-iron complexes (DINICs) are the derivative of nitric oxide (NO), iron, and other ligands. They play a crucial role in stabilization, storage, and NO bio-activity [152]. The electron paramagnetic resonance (EPR) signal at g = 2.03 is a characteristic feature of DINICs and was identified in vivo in animal tissues and other organisms [153]. In general, DINICs are formed by the attachment of anionic ligands to an Fe(NO)_2_ nucleus. One common ligand of these centers is the thiolate of glutathione (GS^−^). Depending on the number of iron-nitrosyl nuclei attached to the ligand(s), both mono- and bi-nuclear DINICs are formed [154]. Under physiological conditions, mononuclear thiol-ligated DINICs appear to be in equilibrium with binuclear DINICs of the Roussin’s red salt thioether type [155]. FeS proteins may be the major source of protein-bound DINICs as demonstrated in *E. coli* when ·NO directly reacts with the FeS clusters [156]. ·NO may also react with superoxide (O_2_^−^), yielding peroxynitrite (ONOO^−^) that can react with proteins, inducing carbonylation and nitration [61]. In a recent study, we provided evidence that FeS-Grx2 can inhibit ONOO^−^ formation in cells by the reaction of its GSH co-ligated FeS cluster with ·NO, yielding glutathionyl-DINICs [88]. Such glutathionyl-DINICs may biologically be the most significant form. For instance, they may play an important role in protein S-nitrosylation (S-NO).

Various studies and reviews have reported the formation of S-NOs in vivo, see for instance [157,158,159,160]. It is often described as the product of the reaction of the ·NO radical with thiol groups. However, this reaction as such cannot take place unless one electron is removed, e.g., by a metal/enzyme catalyst [161].
NO + R-SH → R-S-NO + ***e***^−^(1)

Glutathionyl-DINICs can release nitrosonium ions (NO^+^) that can react with GSH to form GS-NO [162]. This NO^+^ moiety can be transferred to other thiols by a nucleophilic attack on the electrophilic nitrogen atom of GS-NO—a process known as protein trans-nitrosylation [163].
GS-NO + R′-SH → GSH + R′-S-NO(2)

The thiol peroxidases peroxiredoxin 1 (Prx1) is S-nitrosylated in mammalian cells [164]. This reaction is mediated by glutathionyl-DINICs and involves the reactive peroxidatic cysteinyl residue in the active site of Prx1. The reaction affects the reactivity of the protein and thus its enzymatic and signaling functions [165].

DINICs complexed with glutathione may also be of interest for therapeutic applications [153]. They might serve also as ·NO donors to regulate the muscle tonus around the vasculature [152] and have already been successfully tested as hypotensive drugs in clinical trials [166]. The stability of DINICs can be increased through their interaction with proteins [153]. For example, cysteinyl residues in serum albumin can modulate and prolong the vasodilating activity of glutathionyl-DINICs [152]. It is noteworthy that these DINICs do not seem to affect the cellular glutathione levels, nor cellular proliferation [167]. They are not toxic to HeLa cells [168] and they increase the viability of fibroblasts and rat caridiomyocytes [167,169]. Glutathionyl-DINICs were also reported to accelerate skin wound healing, to inhibit apoptosis, and to suppress endometriosis [9,170]. In heart infarction models, treatment reduced the size of the infarction zone and inhibited platelet aggregation [171,172]. In summary, treatment with glutahionyl-DINICS may develop into new therapeutic strategies.

## 4. Conclusions and Outlook

The past decade has seen some remarkable progress in our understanding of the role of GSH and Grxs in iron metabolism, particularly with respect to the synthesis and maturation of FeS proteins. Promising steps have been made towards re-building functional FeS transfer and targeting complexes in vitro. We have reached a molecular understanding of the factors that determine the GSH-dependent oxidoreductase and FeS scaffold functions of Grxs. With the unraveling of the yeast Grx3/4-Fra2-mediated regulation of iron homeostasis in yeast, we have been presented with the first well-characterized physiological function of a Grx-BolA hetero-complex. However, a number of urgent open questions remain to be answered. They include, without claiming completeness:

What is the role of GSH in the export of FeS precursors from mitochondria to the cytosolic FeS assembly machinery? What compound is exported and what is the source of iron for cytosolic FeS maturation?

Outside the well-established yeast Aft regulon, what are the physiological roles of the CGFS-type Grx interactions with BolA-like proteins? Are these interactions essential for mitochondrial and/or cytosolic FeS maturation and transfer reactions? Do (more) of these complexes function in iron sensing?

What is the mechanism of cluster transfer between Grxs and recipient proteins? What are the factors that ensure both the specificity and efficiency of the reactions in vivo that we are apparently missing in vitro to date?

## Figures and Tables

**Figure 1 molecules-25-03860-f001:**
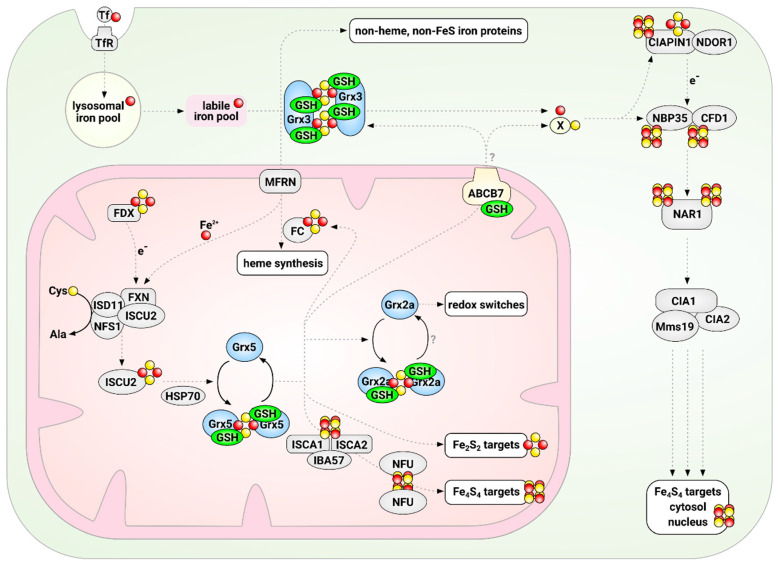
Glutathione and glutaredoxins in iron-sulfur cluster synthesis and maturation in mammalian cells. The initial synthesis of Fe_2_S_2_ clusters is catalyzed by the mitochondrial iron-sulfur cluster synthesis machinery on the scaffold protein ISCU2. From there, clusters are distributed in a process that depends on the CGFS-type Grx5 to Fe_2_S_2_ and Fe_4_S_4_ target proteins, e.g., in the mitochondrial electron chain. In addition, a yet to be uncovered compound “X” is exported in a glutathione (GSH)-dependent manner to the cytosol, where it serves as substrate for the cytosolic iron-sulfur cluster assembly machinery. The multi-domain CGFS-type Grx3 is in some way required for the distribution of iron from the so-called labile iron pool to most, if not all, cellular iron-dependent processes. Glutaredoxins are depicted in light blue, GSH in green, iron in red, and sulfur in yellow.

**Figure 2 molecules-25-03860-f002:**
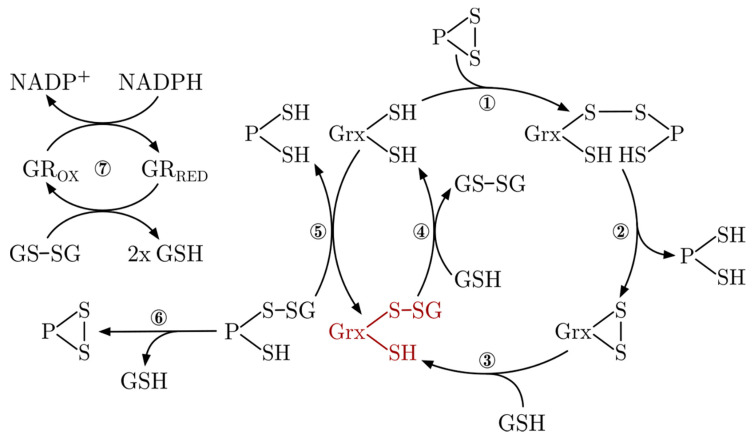
Glutaredoxin reaction mechanisms. Protein disulfides are reduced via a mechanism that involves both active site cysteinyl residues of the CxxC-type Grxs. A reduced Grx forms a mixed disulfide with the thiol of a target protein and its N-terminal active site Cys (1). This intermediate is reduced by the C-terminal active site Cys, releasing the reduced substrate target protein (2). The oxidized Grx can be sequentially reduced by two molecules of GSH (3+4). A Grx-S-SG mixed disulfide (red) can easily be formed from reduced Grx and glutathione disulfide (GSSG) in the reverse reaction (4). Reduced Grx can also catalyze the reversible (de-)glutathionylation of a target protein in a mechanism that only requires the N-terminal active site Cys (5). Some glutathionylated proteins containing two adjacent Cys can also oxidize and form an intra-molecular disulfide by releasing GSH (6). GSSG is reduced to two molecules of GSH by glutathione reductase (GR) at the expense of NADPH.

**Figure 3 molecules-25-03860-f003:**
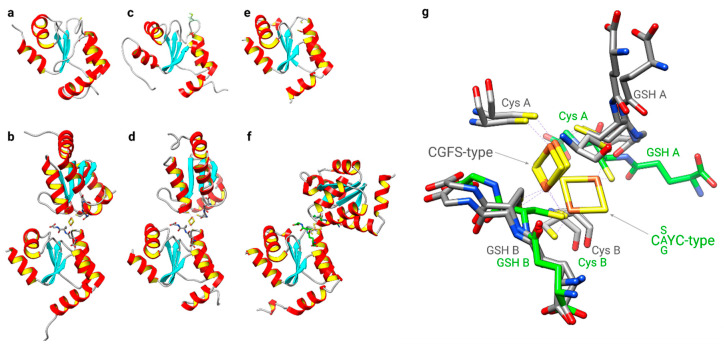
Structural comparison of both classes of FeS glutaredoxins. Structures of (**a**) apo- [PDB:1YKA] and (**b**) dimeric CGFS-type holo-Grx4 [PDB:2WCL] from *Escherichia coli*, CGFS-type human (**c**) apo- [PDB:2MMZ] and (**d**) holo-Grx5 [PDB:2WUL], and CSYC-type human Grx2 in (**e**) apo- [PDB:2FLS] and (**f**) holo-form [PDB:2HT9]. (**g**) Details of the alternative cluster coordination conformations of the holo-complexes, CGFS-type Grxs with GSH and active site cysteinyl residue carbon traces in gray, CSYC-type with carbon traces in green. “A” and “B” refer to the two subunits in the dimeric holo-complexes composed of the two subunits A and B, two non-covalently bound GSHs and the bridging Fe_2_S_2_ cluster.

**Figure 4 molecules-25-03860-f004:**
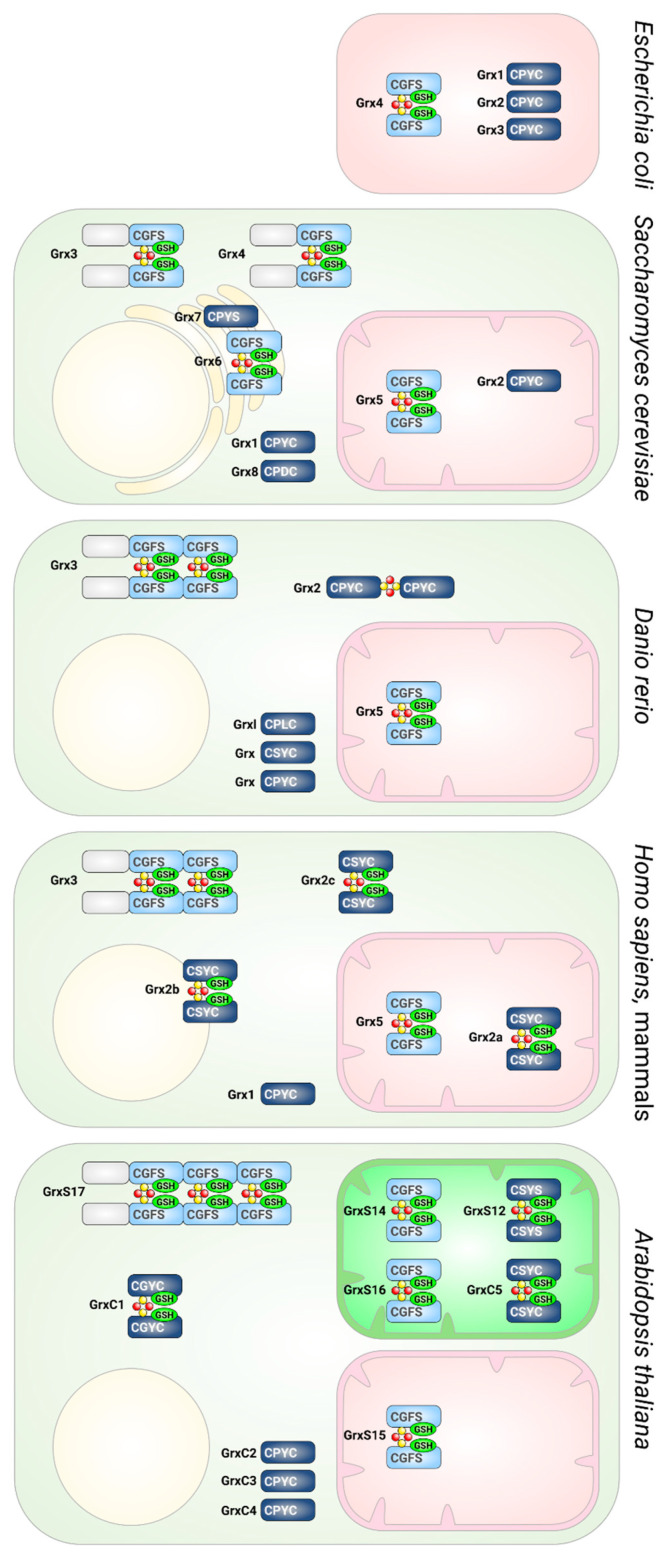
Glutaredoxins in different species. Presence and subcellular localization of Grxs in *Escherichia coli*, *Saccharomyces cerevisiae*, *Danio rerio*, humans, and *Arabidopsis thaliana*. CGFS-type Grxs are depicted in light blue, CxxC-type in dark blue. The domain structures, active site sequences, and the ability to form FeS cluster-bridged holo-complexes are indicated.

**Figure 5 molecules-25-03860-f005:**
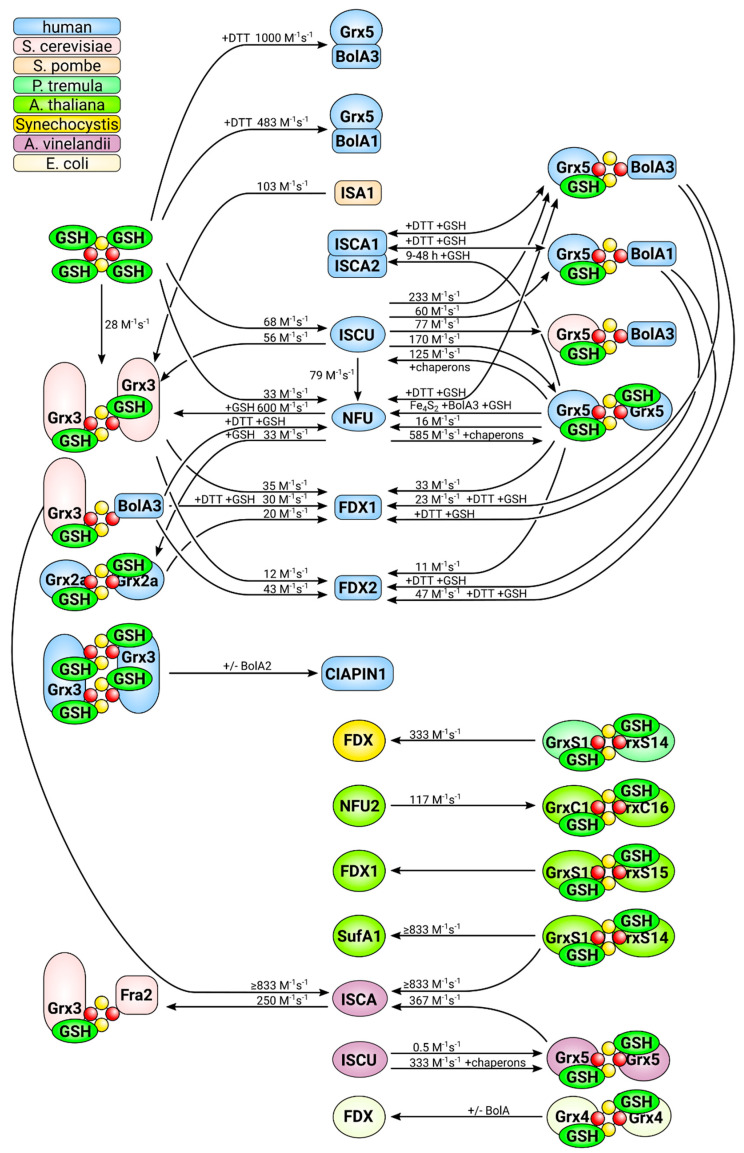
FeS cluster transfer reactions analyzed in vitro to date. This figure summarizes all in vitro cluster transfer reactions analyzed and described in the literature. The directions of the reactions are indicated by arrows. The origin of the different proteins used in the analyses, i.e., the species that encodes the respective protein, is color coded, as displayed in the top left corner. For details and references, see Table 2.

**Figure 6 molecules-25-03860-f006:**
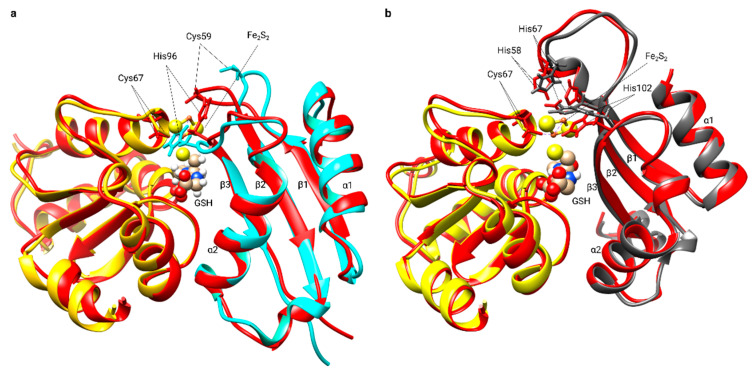
Structural rearrangements in Grx5 and BolA upon dimer formation. (**a**) Superimposition of the backbone structure of human apo-Grx5 [PDB:2WUL] (yellow) and human BolA3 [PDB: 2NCL] (cyan) with the Fe_2_S_2_ BolA3-Grx5 complex backbone structure (red). (**b**) Superimposition of the backbone structure of human apo-Grx5 [PDB:2WUL] (yellow) and human BolA1 [PDB:5LCI] (dark gray) with the Fe_2_S_2_ BolA1-Grx5 complex backbone structure (red). The GSH molecule and the Fe_2_S_2_ cluster are represented as balls and sticks. The invariant C-terminal His (His 96 in BolA3 and His 102 in BolA1), His 67 in BolA1, Cys 59 in BolA3, and Cys 67 in human Grx5 residues are shown.

**Table 1 molecules-25-03860-t001:** Human glutaredoxins. Abbreviations: c: cytosol, m: mitochondria, n.a.: not available, n: nucleus.

Protein Name	Accession	Gene	Active Site	Functions and Reactions	FeS	Locali-Zation
Grx1	P35754	Glrx	CPYC	Oxidoreductase, (de-)glutathionylation [75,76]	no	c/n
Grx2a	Q9NS18	Glrx2	CSYC	Oxidoreductase, FeS as redox sensor [16,77,78]	Fe_2_S_2_	m
Grx2b	Q9NS18-2	Glrx2	CSYC	Not analyzed [77,79]	Fe_2_S_2_	c/n
Grx2c	n.a.	Glrx2	CSYC	Oxidoreductase, FeS as redox sensor [16,77,78]	Fe_2_S_2_	c/n
Grx3	O76003	Glrx3	2·CGFS	Fe/S biogenesis, iron trafficking [52,80]	2·Fe_2_S_2_	c
Grx5	Q86SX6	Glrx5	CGFS	Fe/S biogenesis [81,82]	Fe_2_S_2_	m

**Table 2 molecules-25-03860-t002:** FeS cluster transfer reactions analyzed in vitro. Abbreviations: A.v.: *Azotobacter vinelandii*, H.s.: *Homo sapiens*, P.: *Populus tremodus*, S.c.: *Saccharomyces cerevisiae*, S.p.: *Schizosaccharomyces pombe*, Syn.: *Synechocystis* sp.

Donor	Acceptor	Additional Factors	Transfer Rate (M^−1^ min^−1^)	Method	Assay Time (min)	Ref.
[Fe_2_S_2_](GS)_4_	S.c. Grx3		1360 ± 110 1600 ± 450 2153 ± 281	CD	30	[41]
[Fe_2_S_2_](GS)_4_	H.s. BolA1 H.s. Grx5	DTT	60,000 ± 1200	CD	30	[123]
[Fe_2_S_2_](GS)_4_	H.s. BolA3 H.s. Grx5	DTT	28,000 ± 2800	CD	30	[124]
[Fe_2_S_2_](GS)_4_	H.s. BolA3 S.c. Grx3	DTT	29,000 ± 3000	CD	30	[124]
H.s. Grx2	H.s. Fdx1	GSH	1160 ± 200	CD	60	[122]
H.s. Grx3	H.s. Ciapin1			UV-Vis		[125]
H.s. Grx3	H.s. Ciapin1	BolA2		CD, UV-Vis		[56]
H.s. Grx5	H.s. IscU	DTT, GSH, H.s. HSPA9, HSC20, ATP, MgCl_2_,	7500 ± 2300	CD	60	[115]
H.s. Grx5	H.s. IscU D37A	DTT, GSH	3500 ± 500	CD	58	[115]
H.s. Grx5	H.s. Nfu1	DTT, GSH	950 ± 450	CD	118	[115]
H.s. Grx5	H.s. Nfu1	DTT, GSH, BolA3		CD, UV-Vis, NMR		[126]
H.s. Grx5	H.s. Fdx1	DTT, GSH	2000 ± 700	CD	120	[115]
H.s. Grx5	H.s. Fdx2	DTT, GSH	650 ± 250	CD	120	[115]
H.s. Grx5	H.s. ISCA1/ISCA2	DTT, GSH		NMR	9–48 h	[127]
H.s. Grx5	H.s. ISCA2	DTT, GSH		NMR	9–48 h	[127]
H.s. Grx5	H.s. ISCA2 C79S	DTT, GSH		NMR	9–48 h	[127]
H.s. ISCA1	H.s. BolA1, H.s. Grx5	DTT, GSH		CD	60	[123]
H.s. ISCA2	H.s. BolA1, H.s. Grx5	DTT, GSH		CD	60	[123]
H.s. ISCU	H.s. BolA1, H.s. Grx5	DTT, GSH	3600 ± 400	CD	60	[123]
H.s. ISCU	H.s. BolA3, S.c. Grx3	DTT, GSH	4600 ± 870	CD	60	[124]
H.s. ISCU	H.s. BolA3, H.s. Grx5	DTT, GSH	14,000 ± 1000	CD	60	[124]
H.s. ISCU	H.s. Grx5	DTT, GSH	10,300 ± 1800	CD	10	[115]
H.s. ISCU	S.c. Grx3	GSH	3370 ± 200	CD	60	[128]
H.s. Nfu	H.s. Grx5	DTT, GSH	35,100 ± 2000	CD	60	[129]
H.s. Nfu	H.s. Grx2	GSH	2000 ± 150	CD	60	[122]
H.s. Nfu	S.c. Grx3	GSH	36,200 ± 7700	CD	60	[128]
H.s. Nfu	S.c. Grx3	DTT, GSH, H.s. HSPA9, HSC20, ATP, MgCl_2_	34,400 ± 4500	CD	60	[129]
H.s. Nfu1	H.s. BolA3, S.c. Grx3	DTT, GSH		CD	60	[124]
H.s. Nfu1	H.s. BolA3, H.s. Grx5	DTT, GSH		CD	60	[124]
H.s. BolA1 H.s. Grx5	H.s. Fdx1	DTT, GSH		CD	60	[123]
H.s. BolA1 H.s. Grx5	H.s. Fdx2	DTT, GSH		CD	60	[123]
H.s. BolA1 H.s. Grx5	H.s. ISCA1	DTT, GSH		CD	60	[123]
H.s. BolA1 H.s. Grx5	H.s. ISCA2	DTT, GSH		CD	60	[123]
H.s. BolA3 S.c. Grx3	H.s. Fdx1	DTT, GSH	1800 ± 170	CD	60	[124]
H.s. BolA3 S.c. Grx3	H.s. Fdx2	DTT, GSH	2600 ± 540	CD	60	[124]
H.s. BolA3 S.c. Grx3	H.s. Nfu1	DTT, GSH		CD	60	[124]
H.s. BolA3 H.s. Grx5	H.s. Fdx1	DTT, GSH	1400 ± 140	CD	60	[124]
H.s. BolA3 H.s. Grx5	H.s. Fdx2	DTT, GSH	2800 ± 380	CD	60	[124]
H.s. BolA3 H.s. Grx5	H.s. Nfu1	DTT, GSH		CD	60	[124]
S.c. Grx3	H.s. Fdx1	GSH	2100 ± 500	CD	60	[122]
S.c. Grx3	H.s. Fdx2	GSH	695 ± 138	CD	240	[122]
S.c. Grx3	A.v. IscA		≥50,000	CD	10	[130]
S.p. Isa1	S.c. Grx3	GSH	6200 ± 1900	CD	60	[122]
A.v. Grx5	A.v. Fdx	DTT	2100	CD	160	[131]
A.v. Grx5	A.v. IscA		22,000	CD	10	[130]
A.v. IscA	S.c. Fra2-Grx3		15,000	CD	120	[130]
A.v. IscU	A.v. Grx5	GSH	30	CD	180	[131]
A.v. IscU	A.v. Grx5	A.v. HscA, A.v. HcsB, MgCl_2_, ATP, KCl	20,000	CD	60	[131]
A.t. GrxS14	A.v. IscA		≥50,000	CD	10	[130]
A.t. GrxS14	A.t. SufA1		≥50,000	CD	10	[130]
A.t. GrxS15	A.t. (mito)Fdx1			Native PAGE		[132]
P. GrxS14	Syn. Fdx		20,000	CD	~60	[74]

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
