# Peer review of "Role of GSH and Iron-Sulfur Glutaredoxins in Iron Metabolism—Review"

_molecules, 2020, doi:10.3390/molecules25173860_

Round 1
Reviewer 1 Report
Review paper (ID: molecules-899262) by Daniel et al. is a very interesting study based on the latest bibliographic data on the role of glutathione (GSH) and iron-sulfur glutaredoxins in the regulation of cellular iron metabolism. This relatively poorly known function of GSH is of increasing importance compared to other commonly known functions of this tri-peptide.
This very valuable review work requires some minor corrections. Due to the very complex nature of the reactions presented, it is necessary to introduce abbreviations of individual proteins throughout the text when they appear for the first time (e.g. Glutathione - (GSH) in line 30 instead of 39; iron-sulfur cluster (FeS) containing Grxs (FeS) -Grxs) on line 48 instead of 49).
On line 59, the full name of S. cerevisiae; extension of the IBA57 abbreviation - line 78; Bo1A and NFU1 - line 80; NDUFS1, NDUFV1 -line 82, and all other abbreviations throughout the text.
Author Response
Reviewer 1:
Review paper (ID: molecules-899262) by Daniel et al. is a very interesting study based on the latest bibliographic data on the role of glutathione (GSH) and iron-sulfur glutaredoxins in the regulation of cellular iron metabolism. This relatively poorly known function of GSH is of increasing importance compared to other commonly known functions of this tri-peptide.
This very valuable review work requires some minor corrections. Due to the very complex nature of the reactions presented, it is necessary to introduce abbreviations of individual proteins throughout the text when they appear for the first time (e.g. Glutathione - (GSH) in line 30 instead of 39; iron-sulfur cluster (FeS) containing Grxs (FeS) -Grxs) on line 48 instead of 49).
On line 59, the full name of S. cerevisiae; extension of the IBA57 abbreviation - line 78; Bo1A and NFU1 - line 80; NDUFS1, NDUFV1 -line 82, and all other abbreviations throughout the text.
We thank reviewer 1 for making us aware of this problem. We have thoroughly gone through the whole manuscript and checked all abbreviations again. They are now introduced at the first instance and consequently used throughout the manuscript. Some protein names though, e.g. BolA and IBA57, are generic names and not abbreviations and could therefore not be introduced.
In addition,
1. We have revised the text of the whole manuscript again and corrected a number of small errors.
2. We have changed the order of figures 2 and 3, now they are cited in the text in order of their appearance.
3. We have also included one more just published important reference (Weiler et al, PNAS doi: 10.1073/pnas.2003982117) in the section ‘iron-sulfur cluster biogenesis’:
“Only recently, Lill and co-workers succeeded to biochemically reconstitute the assembly of Fe4S4 clusters 25. The process requires the ISC machinery, holo-Grx5, and reduced Fdx2 for the reductive fusion of two Fe2S2 clusters into one Fe4S4 cluster.”
Reviewer 2 Report
The work is a detailed review of the properties and functions of glutaredoxins focused on their role in the formation and transfer of Fe-S complexes. It is well organized and full of information, although sometimes difficult to follow. It does not give much attention to iron metabolism.
- The focus on iron metabolism is in the title, but not in the text. For example the iron regulatory proteins whose Fe/S clusters have a key role in cellular iron homeostasis are not described and mentioned only once. The title should be changed to a role of glutaredoxins in iron-sulfur metabolism. Alternatively they should produce at least a table that indicates how GSH and Grxs affect iron metabolism.
- I also suggest to include a table that summarizes the properties of the Grxs.
- I am not convinced of the utility of fig 5 that mixes pathways of pro- and eu-karyotes of 8 species. A simplified version with a legend that explains its significance would be more useful.
there are a few typos and errors
- page 1, line 20 mutli-domain
- p 3 l 90, the sentence misses the verb
- p 3, l 111, see x, misses the reference
- p 4 l 132, two FeS cluster, misses the s
- p 10, l 307, the sentence misses “was”
- p 11, l 366, the sentence is very generic and obscure. What iron-depending means in this contest?
Author Response
Reviewer 2:
The work is a detailed review of the properties and functions of glutaredoxins focused on their role in the formation and transfer of Fe-S complexes. It is well organized and full of information, although sometimes difficult to follow. It does not give much attention to iron metabolism.
- The focus on iron metabolism is in the title, but not in the text. For example the iron regulatory proteins whose Fe/S clusters have a key role in cellular iron homeostasis are not described and mentioned only once. The title should be changed to a role of glutaredoxins in iron-sulfur metabolism. Alternatively they should produce at least a table that indicates how GSH and Grxs affect iron metabolism.
We acknowledge reviewer 2’s concern. However, FeS and Fe metabolism in general are inseparable. In the previous draft, our manuscript already included sections and references describing the effect of FeS-Grxs and GSH on the regulation of iron metabolism in yeast, as well as the GSH-including di-nitrosyl iron complexes. In addition, we have amended the following sections to the effect of the Grxs on the iron regulatory protein (IRP) 1 and iron metabolism in general, in their respective sections:
Grx2: “In models of Parkinson’s disease, depletion of glutathione resulted in a dose-dependent Grx2 inhibition and, similar to gene silencing of Grx2, decreased iron incorporation into complex I and Aco2. Loss of Grx2 function also led to the activation of iron-regulatory protein (IRP1), resulting in the increase in the iron uptake protein transferrin receptor, decreased levels of the iron storage protein ferritin, and mitochondrial iron accumulation. In the cytosol, the loss of Grx2 resembled iron-starvation conditions.”
Grx3: “Gene silencing of Grx3 in cells of human origin (HeLa cells) induced a phenotype resembling an iron starvation phenotype despite of sufficient bio-available iron. Protein levels of several cytosolic FeS-proteins were altered, for instance IRP1 and GPAT. The protein levels of ferritin were decreased, the levels of the transferrin receptor increased, indicating the activation of IRP1. Apparently, the Grx3-depleted cells were unable to use iron efficiently, indicating a central role for Grx3 in iron metabolism”
Grx5: “In zebrafish, lack of Grx5 led to the activation of IRP1 and blocked heme biosynthesis. The first step in this pathway is catalyzed by aminolaevulinate synthase 2 (ALAS2). Over-expression of ALAS2 RNA without the iron response element regulated by IRP1 rescued the zebrafish embryos, while the expression of ALAS2 including the iron response element did not. Human patients with decreased levels of Grx5 develop an iron overload and sideroblastic-like microcytic anemia.”
- I also suggest to include a table that summarizes the properties of the Grxs.
As requested, we have included a new table (table 1) that summarizes the characteristics and properties of the human Grxs.
- I am not convinced of the utility of fig 5 that mixes pathways of pro- and eukaryotes of 8 species. A simplified version with a legend that explains its significance would be more useful.
We apologize that our intention for Fig.5 was not made clear enough before. The figure summarizes (to the best of our knowledge) all in vitro analysed cluster transfer reactions published until today. This figure was explicitly added to allow the unbiased readers to evaluate the significance of the published in vitro transfer reactions themselves. This is an important part of our critics on all these published reactions. We have thus opted to leave Fig. 5 unaltered in the manuscript. We agree, however, that the figure legend needed to be improved. It now reads:
“Figure 5 – FeS cluster transfer reactions analyzed in vitro until today.
This figure summarizes all in vitro cluster transfer reactions analyzed and described in the literature. The direction of the reactions are indicated by arrows. The origin of the different proteins used in the analyses, i.e. the species that encodes the respective protein, was color-coded as displayed in the top left corner. For details and references, see also table 1.”
there are a few typos and errors
- page 1, line 20 mutli-domain- p 3 l 90, the sentence misses the verb
- p 3, l 111, see x, misses the reference
- p 4 l 132, two FeS cluster, misses the s
- p 10, l 307, the sentence misses “was”
- p 11, l 366, the sentence is very generic and obscure. What iron-depending means in this contest?
We thank reviewer 2 for identifying theses errors. We have corrected them and rephrased the sentence on p11 as follows:
“The common thread in these regulatory mechanisms are the multi-domain CGFS-type Grxs that interact, often together with BolA-type proteins (see below) with transcription factors dependent on the iron state of the cell thus controlling the transcription of proteins and enzymes that take part in, or control iron metabolism.”
In addition,
1. We have revised the text of the whole manuscript again and corrected a number of small errors.
2. We have changed the order of figures 2 and 3, now they are cited in the text in order of their appearance.
3. We have also included one more just published important reference (Weiler et al, PNAS doi: 10.1073/pnas.2003982117) in the section ‘iron-sulfur cluster biogenesis’:
“Only recently, Lill and co-workers succeeded to biochemically reconstitute the assembly of Fe4S4 clusters 25. The process requires the ISC machinery, holo-Grx5, and reduced Fdx2 for the reductive fusion of two Fe2S2 clusters into one Fe4S4 cluster.”